# Targeted and Non-Targeted Mechanisms for Killing Hypoxic Tumour Cells—Are There New Avenues for Treatment?

**DOI:** 10.3390/ijms22168651

**Published:** 2021-08-11

**Authors:** Alyssa Gabrielle Apilan, Carmel Mothersill

**Affiliations:** Department of Biology, McMaster University, Hamilton, ON L8S 4L8, Canada; mothers@mcmaster.ca

**Keywords:** hypoxia, non-targeted effects, autophagy, PET imaging, radiosensitizers

## Abstract

Purpose: A major issue in radiotherapy is the relative resistance of hypoxic cells to radiation. Historic approaches to this problem include the use of oxygen mimetic compounds to sensitize tumour cells, which were unsuccessful. This review looks at modern approaches aimed at increasing the efficacy of targeting and radiosensitizing hypoxic tumour microenvironments relative to normal tissues and asks the question of whether non-targeted effects in radiobiology may provide a new “target”. Novel techniques involve the integration of recent technological advancements such as nanotechnology, cell manipulation, and medical imaging. Particularly, the major areas of research discussed in this review include tumour hypoxia imaging through PET imaging to guide carbogen breathing, gold nanoparticles, macrophage-mediated drug delivery systems used for hypoxia-activate prodrugs, and autophagy inhibitors. Furthermore, this review outlines several features of these methods, including the mechanisms of action to induce radiosensitization, the increased accuracy in targeting hypoxic tumour microenvironments relative to normal tissue, preclinical/clinical trials, and future considerations. Conclusions: This review suggests that the four novel tumour hypoxia therapeutics demonstrate compelling evidence that these techniques can serve as powerful tools to increase targeting efficacy and radiosensitizing hypoxic tumour microenvironments relative to normal tissue. Each technique uses a different way to manipulate the therapeutic ratio, which we have labelled “oxygenate, target, use, and digest”. In addition, by focusing on emerging non-targeted and out-of-field effects, new umbrella targets are identified, which instead of sensitizing hypoxic cells, seek to reduce the radiosensitivity of normal tissues.

## 1. Introduction

### Background to Tumour Hypoxia

One of the most difficult barriers encountered when treating the majority of solid tumours is attributed to the scattered microregions within the tumour characterized by the lack of oxygen. This is known as tumour hypoxia [1]. In order to maintain functionality within normal tissues, such as excreting metabolic waste, oxygen must be supplied through existing vasculature systems [2,3,4]. During the early stages of tumourigenesis, prior to rapid tumour cell proliferation, tumours use existing vasculature systems for oxygen supply in order to maintain the same metabolic demands as normal tissues [2,5,6]. However, as tumour development continues, these cells begin to proliferate rapidly, leading to a sharp increase in size and mass [1,7,8]. Consequently, rapid tumour growth results in an increased demand for oxygen to supply the tumour cells; however, the body does not have the adequate oxygen supply to meet the heightened demands [1,7,8]. Furthermore, as the tumour continues to enlarge, there is an increase in the distance between tumour cells and existing vasculature [1,7,8]. In response to the increased distance, tumours will use the existing vasculature to develop an independent vasculature system through the process of angiogenesis [2,5,6]. However, the newly formed vasculature is severely flawed both on a structural and functional level, and this impairs the supply of oxygen to tumour cells that require oxygen to satisfy metabolic demands [2,3,4]. Moreover, the lack of oxygen supply results in hypoxic microregions scattered throughout the tumour relative to normal tissue [1]. The hypoxic environments within tumours create several cancer treatment barriers. Most notably, all hypoxic cells are resistant to ionizing radiation (IR) [9,10,11]. The mechanism by which radiation is able to eradicate tumour cells by damaging DNA, resulting in apoptosis and cell death, occurs through the production of reactive oxygen species (ROS) [12,13]. However, due to the limited oxygen availability within hypoxic tumour microenvironments, this impedes the efficacy of radiotherapy [12,14]. Furthermore, normal tissues are unable to withstand increased doses of radiation that compensate for tumour hypoxia [12]. As tumour hypoxia and normal tissue radiation dosage limitations have prevented effective radiotherapy, Hall and Giaccia [15] have outlined four methods targeting tumour hypoxia through radiosensitizing hypoxic cells. These include hyperbaric oxygen, hypoxic cell radiosensitizers, hypoxic cytotoxins, and tumour metabolism. However, as research continues to elucidate the relationship between tumour hypoxia and radiotherapy, novel approaches have been developed. This review outlines four emerging approaches for targeting and radiosensitizing hypoxic cells, namely, guided carbogen breathing through tumour hypoxia imaging (i.e., PET), which increases the oxygen levels in targeted hypoxic areas, gold nanoparticles, macrophage-mediated drug delivery, and autophagy inhibitors, which result in the removal of hypoxic cells. Particularly, these novel methods improve the therapeutic ratio between hypoxic tumour microenvironments and normal tissue, thus, increasing the efficacy of anticancer therapeutics. The review also focuses on non-targeted effects (NTE) and considers the possibility that rather than trying to sensitize hypoxic cells, we could try to protect normal tissues from low-dose collateral effects by inhibiting or reducing bystander effects, which signal p53 competent cells to undergo apoptosis or other forms of cell death [16,17]. The novel methods and targets are shown in graphic form in Figure 1.

## 2. Current Approaches to Radiosensitizing Hypoxic Tumour Cells

### 2.1. Tumour Hypoxia Imaging via PET for Guiding Carbogen Breathing Therapy

#### 2.1.1. Background

One of the earliest techniques used to control tumour hypoxia was hyperbaric oxygen (HBO). Figure 2 shows the rationale behind the modern approach to sensitize hypoxic cells by identifying, imaging, and measuring oxygen levels in hypoxic areas to improve management. The intent of HBO treatment is to increase the supply of oxygen to hypoxic tumour cells. When this treatment is coupled with radiotherapy, this allows the reoxygenated hypoxic tumour cells to become more radiosensitive, thus, reducing the progression of metastasis [18,19,20]. However, due to patient concerns associated with claustrophobia, the time needed for the administration of the treatment, and equal effectiveness using drugs, has led to a shift from hyperbaric oxygen [15,21]. In addition, a systematic review conducted by Bennett et al. [22] suggests more side effects associated with HBO, such as oxygen poisoning and severe radiation injury. Despite the adverse effects associated with HBO, previous studies present conflicting evidence. Particularly, a study conducted by Kohshi et al. [23] suggests that radiotherapy should be performed immediately after HBO treatment, rather than the two procedures occurring simultaneously, to avoid adverse effects from HBO. Based on these conclusions, the use of HBO for treating tumour hypoxia continues to be controversial [24].

The uncertainty surrounding HBO treatment for combating tumour hypoxia has shifted to using a combination of 5% CO_2_ and 95% O_2_ to create a gaseous compound known as carbogen [25]. Recent research has shifted away from solely using oxygen manipulation-based methods to guiding carbogen therapy through medical imaging, specifically PET, to predict hypoxic regions of a tumour in order to alter tumour hypoxia therapeutics accordingly [26,27]. Although research has been conducted using various imaging techniques, PET imaging is preferred due to high precision and sensitivity in vivo, as well as providing measurements of intracellular oxygen levels [28]. Moreover, tumour hypoxia imaging using PET allows for identifying novel indicators of tumour hypoxia, as well as aiding in determining baseline responses elicited from hypoxic tumours following hypoxia therapeutics [28]. To facilitate the identification of hypoxic regions within a tumour through PET imaging, a PET radiotracer that is suitable for all types of cancer must be used [28]. However, the most appropriate radiotracer has yet to be identified, but research continues to examine novel and existing compounds in relation to tumour hypoxia imaging and hypoxia therapeutics [28]. Despite this approach being in the early stages, tumour hypoxia imaging can serve as a powerful tool to identify and treat hypoxic tumour microenvironments for cancer therapies relative to normal tissues.

#### 2.1.2. Mechanism of Action

As previously stated, in order to facilitate effective tumour hypoxia imaging, an “ideal” PET radiotracer is required and must meet a number of biochemical characteristics [28]. Although many compounds are being investigated, virtually all compounds do not meet all the criteria of the “ideal” PET tracer, nor are they available for imaging all tumour types [29]. Despite the lack of an “ideal” PET tracer, research has been focused on nitroimidazole analogs, specifically, 2-nitroimidazole [28]. Although nitroimidazoles were originally intended to be radiosensitizers, Chapman et al. [30] demonstrated that these compounds can serve as hypoxia markers. These compounds are able to passively diffuse into cells, which is largely determined by the intracellular environment [31]. The main driver of the initial reduction following passive diffusion of nitroimidazoles is the concentration of intracellular oxygen [31]. Once the compounds have entered the cell, nitroimidazoles will undergo single-electron reduction to create a free radical anion [31]. Subsequently, within normoxic cells, free radical anions are promptly reoxidized to the parent compound through intracellular oxygen levels due to the high electron affinity relative to the nitro group on nitroimidazole [31]. In contrast, following single-electron reduction of nitroimidazole within hypoxic tumour cells, due to low intracellular oxygen concentrations, reoxidation cannot be completed [31]. Subsequently, incomplete reoxidation results in the further reduction of the free radical anion, creating a very reactive species that are able to bind to components of a cell [31,32]. Furthermore, reduced nitroimidazole has been shown to accumulate within hypoxic cells, thus, demonstrating its potential as a PET tracer [28]. Of the nitroimidazole analogs screened as PET tracers, most compounds are fluorinated nitroimidazoles; however, ^18^F-fluoromisonidazole (^18^F-FMISO) has garnered the most success and has been extensively studied [28]. The mechanism of nitroimidazole analogs entrapment within hypoxic tumour cells can be applied to ^18^F-FMISO. ^18^F-FMISO is a lipophilic compound; thus, it is readily available to passively diffuse into cells, subsequently, reduction of this compound via the nitroreductase enzyme (NTR), results in the production of R-NO_2_ radicals [33]. Furthermore, due to the low intracellular oxygen levels (pO_2_ < 10 mmHg), these radicals are unable to be reoxidized, leading to further reduction of R-NO_2_ radicals to form R-NHOH molecules that can bind to cellular components, allowing for tumour hypoxia imaging [33,34,35,36]. Recently, another type of fluorinated nitroimidazole, ^18^F-Fluoroazomycin arabinoside (^18^F-FAZA), has been gaining more popularity relative to ^18^F-FMISO [33]. Studies suggest that ^18^F-FAZA, in comparison to ^18^F-FMISO, is less lipophilic due to the presence of an additional sugar moiety [33,37,38]. Moreover, due to structural composition differences,^18^F-FAZA has a faster diffusion and clearance rate relative to ^18^F-FMISO, allowing for an improved tumour-to-background (T/B) ratio, thus, this compound has been gaining more interest as a PET radiotracer [39,40]. Overall, due to the new spotlight on ^18^F-FAZA as a PET radiotracer for tumour hypoxia imaging, researchers are beginning to investigate the use of PET imaging to mediate oxygen-based manipulation therapies, such as carbogen breathing. Thus, tumour hypoxia imaging serves as a promising method to identify hypoxic tumour microenvironments relative to normal tissue, ultimately improving the efficacy of oxygen-based manipulation therapies. In relation to the non-targeted effect theme of this review, which is discussed in detail later, it is likely that increased oxygen will increase oxidative stress and lead to initiation of both bystander signalling and genomic instability phenotypes. To what extent this impacts the therapeutic ratio is not known and is an area for further research.

#### 2.1.3. Current Studies and Future Considerations

Due to the ongoing barrier of tumour hypoxia causing resistance to IR, there has been an increase in research regarding tumour hypoxia imaging using PET scans in order to modify treatment methods, such as carbogen breathing [26]. As previously outlined, ^18^F-FAZA has been gaining recent attention as a PET radiotracer, in relation to the popular counterpart, ^18^F-FMISO, due to increased efficacy in imaging. An experiment conducted by Tran et al. [26] explored the use of tumour hypoxia imaging, particularly with PET imaging in tandem with the PET radiotracer, ^18^F-FAZA PET, as a tool for guiding treatments that target tumour hypoxia, such as carbogen breathing. In this study, Tran et al. [26] used two hypoxic rat tumour models, rhabdomyosarcoma and 9L-glioma, and all were injected with 18.5–25.9 MBq ^18^F-FAZA with a volume of 500 μL. Within both tumour models, a control group was set aside and was not subjected to IR; in addition, rats with 9L-glioma were exposed to 40 Gy [26]. In contrast, rats with rhabdomyosarcoma that were not part of the control group were exposed to either 15 or 20 Gy of IR. Additionally, in order to analyze the efficacy of carbogen on tumour hypoxia, PET imaging with ^18^F-FAZA PET was carried out prior to irradiation. Moreover, treatment groups (i.e., rat groups subjected to IR) were randomly organized into two distinct groups, breathing in air or carbogen [26]. Rats subjected to carbogen breathing were encased within a chamber of carbogen at the flow rate of 2 L/min, 30 min prior to being injected with ^18^F-FAZA, and stayed under these conditions until the end of the study [26]. Subsequently, PET imaging occurred three hours after ^18^F-FAZA administration [26]. In addition, in order to measure the intracellular oxygen levels (pO_2_) within tumours, EPR oximetry was used. Using this approach, it was confirmed that both rat models displayed pO_2_ < 10 mmHg confirming that they are hypoxic [26]. In order to determine if there was an effect on tumour growth, rats were anesthetized, and researchers used caliper measurements at the start of treatment (D0) and 10 weeks after treatment to analyze if there was tumour growth [26]. A previous study conducted by Tran et al. [41] has linked pO_2_ levels less than 10 mmHg to a T/B ratio greater than 1.7 when using ^18^F-FAZA, whereby hypoxic tumour microregions remained radioresistant. The significance of 10 mmHg serves as a threshold for distinguishing between radioresistivity (i.e., pO_2_ < 10 mmHg) or radiosensitivity (i.e., pO_2_ > 10 mmHg). Furthermore, a T/B ratio of 1.7 serves as a benchmark to determine the efficacy of radiotherapy following tumour hypoxia therapeutics such as carbogen breathing [26]. Moreover, the group analyzed the response of the two tumour models to carbogen breathing using ^18^F-FAZA using PET imaging and EPR oximetry [26]. Based on the PET imaging results, Tran et al. [26] produced statistically significant results (*p* < 0.05) for both tumour models; however, rats with 9L-gliomas experienced a greater response to carbogen breathing relative to rats with rhabdomyosarcoma. Furthermore, through EPR oximetry measurements, rats with 9L-gliomas demonstrated a greater increase in intracellular levels of oxygen following carbogen breathing in comparison to rats with rhabdomyosarcoma, which did not surpass the threshold of pO_2_ for displaying radiosensitivity [26]. Moreover, Tran et al. [26] analyzed the effect of irradiation following carbogen breathing to determine if there was the effect of tumour growth delays. Tran et al. [26] have determined statistically significant results for tumour growth delays within rats with 9L-gliomas but not in rats with rhabdomyosarcoma. From this study, it can be concluded that ^18^F-FAZA serves as an attractive PET tracer for PET tumour hypoxia imaging in order to dictate oxygen manipulation-based therapeutics such as carbogen breathing. In addition, the results from this study display promising evidence that tumour hypoxia imaging via PET imaging can potentially become a standardized diagnostic approach for identifying hypoxic tumour microenvironments in order to improve anticancer therapies. Moreover, future studies should analyze the efficacy of using ^18^F-FAZA with PET for tumour hypoxia imaging in order to measure the effectiveness of anticancer therapies outside of oxygen manipulation-based therapies. Overall, although this method has yet to be translated to a clinical setting and other forms of cancer, this study serves as an exciting gateway for further exploring the role of ^18^F-FAZA as a PET tracer and its connections to mitigating tumour hypoxia, and as a novel approach to better detect hypoxic tumour sites.

### 2.2. Gold Nanoparticles

#### 2.2.1. Background

Due to the concerns associated with HBO treatment, researchers began to investigate compounds that mimic oxygen, which also allows for the radiosensitization of hypoxic tumour cells. Figure 3 shows the concept behind the modern use of novel hypoxic cell sensitizers. The use of chemical compounds led to the development of a group of compounds classified as nitroimidazoles [42]. Particularly, these groups of drugs are able to differentiate between normal tissue and tumours due to the lack of intracellular oxygen in hypoxic cells [43,44]. Of the four compounds identified, nimorazole was identified as being the least active but displayed the most effectiveness, particularly within patients with head and neck cancers [45,46]. Due to the primitive tumour vasculature system, a diffusion barrier is created between the tumour cells and the blood vessels carrying anticancer drugs, thus, limiting the effectiveness of nitroimidazoles [12,47]. Despite the inviting potential of these drugs, high dosages induce neurotoxic effects on the central nervous system [48]. In addition, Wardman [48] states that some types of nitroimidazoles may reduce the concentration of thiol within normal tissue. Thiols are known to be radioprotective compounds; thus, the depletion of these molecules within hypoxic tumour cells will induce radiosensitization [48]. Consequently, a reduction in thiol concentrations within normal tissue can potentially subject these tissues to radiosensitization, hence, posing detrimental problems [48]. Current research has begun to use nanotechnology to develop novel cancer radiosensitizers comprised of metallic nanomaterials [49]. The integration of nanotechnology within tumour hypoxia therapy is aimed at enhancing the enhanced permeability and retention (EPR) effect, which is a biological dysfunction characteristic of tumours [50]. Due to the novel vasculature system fabricated within tumours, the rapid rate of proliferation leads to the generation of local compressive forces within the vasculature system [51,52]. In particular, the compressive force on the lymphatic vessels results in reduced lymphatic drainage [51]. In addition to the EPR effect, these novel tumour vasculature systems, relative to normal vasculature systems, display larger pores [53]. Due to the large fenestrations, this allows nanomedicines to more easily enter tumours; furthermore, owing to the poor lymphatic drainage systems, nanomedicines that have entered are able to accumulate and take effect within hypoxic tumour regions [53,54,55,56,57]. The primitive characteristics of the tumour-generated vasculature systems are collectively known as the EPR effect and are the target for successful drug delivery [51,58,59]. Wang et al. [49] have indicated that nanomaterials with a high atomic number (Z) are the most promising radiosensitizers due to manufacturing feasibility, size, energy absorption, as well as scattering and emission of radiation energy. In addition, metallic nanomaterials such as gold and silver have demonstrated low toxicity, fast distribution, and agreeable kinetic profiles [49,60,61,62]. Of the metallic nanomaterials, gold nanoparticles (GNPs) demonstrate the most promise due to strong photoelectric absorption, impeccable biocompatibility, and low toxicity [63]. Furthermore, GNPs have a large volume to surface area ratio allowing for other therapeutics to be used, increased effect on EPR, low permeability, contrasting ability in imaging technology, and controlled size distribution [63]. Hence, GNPs themselves and novel drug delivery systems demonstrate promising capabilities as novel radiosensitizers in order to improve targeting to hypoxic tumour sites relative to normal tissue and to decrease toxicity.

#### 2.2.2. Mechanism of Action

Aside from the classic compounds used as radiosensitizers, GNPs have demonstrated promising results as novel hypoxic cell radiosensitizers. Following exposure to IR within hypoxic cells subjected to GNPs, these materials undergo three distinct interactions with IR, physical, chemical, and biological, to induce radiosensitization [60]. First, within nanoseconds of IR exposure, interactions on a physical level begin radiosensitizing hypoxic tumour cells [60]. Due to the discrepancy in energy absorption abilities between gold and soft tissues, gold is an attractive material that can be used to induce physical dose enhancement [60]. Primarily, there are two main mechanisms whereby photons lose energy, namely, the Compton effect and the photoelectric effect [63]. The Compton effect is characterized by the scattering of incident photons caused by colliding with electrons that are weakly held [63]. In addition, during the collision between the incident photon and weakly bounded electrons, the energetic photons will transfer some energy to the electrons, causing the ejection of electrons from the atom [63]. Furthermore, Chen et al. [63] state that in events where the Compton effect is dominant, despite the small amounts of energy transfer, photons retain the majority of the energy and decelerate over long ranges, thus, exhibiting sparse areas of ionization. Contrastingly, the photoelectric effect is distinguishable from the Compton effect because of the strong dependency between the photon energy and electron binding energy [63]. As a result of this dependency, when an incident photon is absorbed by an electron bound to an atom, this leads to the ejection of an inner bound electron [60]. Furthermore, Her et al. [60] explain that in order to compensate for the ejection of an inner bound electron, electrons situated on the outer-shell fall inwards, simultaneously, causing lower energy photons (fluorescence) and a variety of secondary electrons, known as Auger electrons, to be released. Moreover, in order to radiosensitize hypoxic cells, GNPs take advantage of the atom number discrepancy between the high atomic number of gold (Z = 79) and the low atomic numbers of soft tissues [60]. Overall, Her and colleagues [60] state that the difference in atomic numbers allows GNPs to deliver more energy per unit mass, hence, leading to the increased local deposition of radiation within hypoxic areas of a tumour [60].

Following physical interactions with GNPs, chemical interactions soon take effect. Although the mechanisms underlying these interactions have not been completely elucidated, studies suggest that chemical enhancement occurs through two different pathways [60]. The first pathway suggests that DNA becomes chemically sensitized following IR-induced damage, while the secondary pathway suggests that the active surface of GNPs causes the increased formation of radicals and catalysis, leading to chemical sensitization [60]. Chemical sensitization of DNA occurs through the nuclear localization of GNPs to bind to DNA, which causes chromatin structures to “open”, thus, increasing DNA sensitivity to IR [63]. Moreover, electrons with an ionization threshold of <10 eV, known as low energy electrons (LEEs), and secondary electrons are critical for the radiosensitization process [63,64,65]. Although the interaction between LEEs and GNPs does not produce secondary electrons, Chen et al. [63] argue that significant DNA damage can be inflicted. Furthermore, Chen et al. [63] suggest that LEEs cause transient negative ions to weaken the hydrogen bonds within DNA, thus increasing chemical sensitivity. However, it is critical to be cognizant of the charge and size of GNPs, since chemical sensitization depends on these characteristics [60,63,65]. The latter mechanism of DNA chemical sensitization is attributed to the activated surfaces of GNPs, which catalyze a variety of chemical reactions [60,63,66,67]. Specifically, attention has been focused on GNPs (<5 nm) with large surface areas, which demonstrate large catalytic activity governing the transfer of electrons from surface-bound donor groups to O_2_ to produce free radicals [63]. According to Her et al. [60], due to the small size and curved structure of nanoparticles, this destroys the impeccable structure and organization of gold to produce free radicals on GNPs. Alternatively, the catalytic reactions induced by GNPs can lead to the transfer of electrons and increased production of ROS [63]. Moreover, increased levels of ROS induce negative implications on the biological interactions between GNPs and IR, particularly through oxidative stress [63].

The biological interactions between GNPs and IR occur through three different pathways, oxidative stress, disruption of the cell cycle, and inhibition of DNA repair [60]. One of the primary pathway’s radiation can induce cell killing is through the radiolysis of water, which generate free radicals, and allows ROS to interact with other components of the cell [60]. As described by Her et al. [60], ROS, superoxide radicals (O_2_^−^), hydroxyl radicals (OH), and hydrogen peroxide (H_2_O_2_) interact with cellular components to induce cellular damage through two different mechanisms. Her and colleagues [60] state that the aforementioned molecules can have direct actions with cell components to directly or indirectly induce oxidative stress, which ultimately triggers cell death through necrosis or apoptosis. Thus, the increased production of ROS mediated by GNPs leads to cell damage through increased oxidative stress, which is the primary characteristic of nanoparticles inducing cytotoxicity [60]. Although the underlying cellular mechanisms are not well understood, recent studies suggest that mitochondria also amplify ROS production [63]. Several groups [68,69] suggest that oxidative stress can induce mitochondrial dysfunction, whereby a multitude of biological effects can lead to apoptosis or necrosis. Chen et al. [63] indicate that GNP-driven oxidative stress leading to increased production of ROS and is linked to mitochondrial dysfunction, which can potentially lead to heightened cell death. Despite the favourable observations, further research must be conducted to elucidate and validate the role of mitochondria and ROS production. Secondly, GNPs disrupt the cell cycle. Within mammalian cells, IR is known to halt cells within the G1 or G2 phase [59]. Moreover, the stages of the cell cycle exhibit various levels of radiosensitivity, whereby cells within the late S-Phase display the maximum radioresitivity, while cells in the G2/M phase are most the radiosensitive [70]. Mackey et al. [71] have elucidated that GNPs have the ability to alter cell-cycle distribution, such that there is an increase in cells within the G2/M phase, hence, increasing radiosensitivity. However, conflicting results from several studies [63,72,73,74] indicate that GNPs do not have an influence on cell-cycle distribution. Evidently, due to the conflicting conclusions between GNPs and cell-cycle distribution, this relationship must continue to be investigated. Finally, IR is known to inflict a variety of DNA damages, namely, double-stranded breaks (DSBs), single-stranded breaks (SSBs), DNA-protein crosslinks, and modifications to DNA bases; however, DSBs are the most lethal [63]. The inability to repair DSBs causes a cascade of cellular impairments that ultimately lead to cell death and can occur in a multitude of ways [63]. Using comet assays in tandem with biomarkers sensitive to DNA damage, such as phosphorylated histone variant γ-H2AX and p53-binding protein 1 (53BP1), this can be employed to uncover the effect of GNPs on DNA repair following exposure to IR [63,75,76,77]. Chen et al. [63] state that several dynamic monitoring experiments using γ-H2AX and 53BP foci assay have been able to detect DNA damage, following the employment of GNPs and IR. Although GNPs’ impact on DNA damage serves as a plausible pathway for radiosensitization, earlier studies have presented conflicting results; thus, further investigations must be conducted to validate the connection [63]. Overall, the physical, chemical, and biological interactions of GNPs leading to radiosensitization have not been completely uncovered. Significantly, no research appears to have been performed looking at the possible involvement of NTE. Metals and other inorganic and organic chemicals are known to produce NTE and to increase genomic instability and bystander effects in vitro and in vivo [78,79,80,81,82,83]. It is probably important to consider whether the mechanisms by which GNPs lead to radiosensitization involve the induction of NTE.

Furthermore, research has also been conducted on developing various methods for nano-drug delivery systems (NDDS) in order to effectively deliver radiosensitizers to hypoxic regions of a tumour [84]. One such example of an NDDS to deliver radiosensitizers to hypoxic tumour microenvironments include GNPs encased in nanoshells fabricated from titanium oxide, allowing for the radiosensitization by halting the cell cycle and increasing oxygenation through the deposition of titanium oxide [49]. In addition, alternative NDDS methods used for radiosensitizers include mesoporous silica nanoparticles, bovine serum albumin proteins nanocapsules, liposomes, nanostructured lipid carriers, and lipid nanocapsules [49]. Thus, NDDS is an attractive method for targeting hypoxic tumour microenvironments relative to normal tissues in order to effectively deliver GNPs. However, subsequent experiments must be conducted to elucidate the biological mechanisms and delivery methods in order to validate and potentiate GNPs as novel radiosensitizers for hypoxic cells. Hence, the simultaneous employment of GNPs and NDDS serves as an attractive approach for surpassing the shortcomings of nitroimidazoles, namely, radiosensitization of normal tissue and improving radiotherapy.

#### 2.2.3. Current Studies and Future Considerations

Despite the high potential of GNPs as novel radiosensitizers, these compounds have not yet been approved for clinical use. However, several clinical trials have demonstrated the efficacy and potential of using these compounds in tandem with radiotherapy to treat tumour hypoxia [63]. Chen et al. [63] highlight four notable clinical trials that have demonstrated compelling evidence of the use of GNPs as novel radiosensitizers. Koonce and colleagues [85] conducted phase 0 and 1 trials using GNPs coated with pegylated recombinant human tumour necrosis factor, collectively classified as CYT-6091, on various stages of tumour development. Similarly, a study funded by Nanospectra Biosciences has developed AuroShell^®^, which is comprised of pegylated silica core-gold shell nanoparticles used within photothermal therapy for head and neck cancers [63]. Kumthekar and colleagues [86] conducted a phase 1 trial using GNPs coated with nucleic acids, known as NU-0129 GNPs, to treat patients with gliosarcoma or recurrent glioblastoma multiforme. Finally, Khoobchandani and colleagues [87] have completed clinical trials using a GNP-based drug known as Nano Swarna Bhasma. The results produced from Khoobchandani and colleagues [87] have provided significant results within women diagnosed with IIIA or IIIB breast cancer and have been approved for clinical use by the Indian government.

Overall, all four studies demonstrate the potential for GNPs as novel radiosensitizers; however, there is a general consensus that many questions regarding the translation of GNPs to a clinical setting have not yet been answered. A review by Schuemann et al. [88] suggests that three major areas must be thoroughly considered prior to clinical applications, specifically, optimal GNP structure and design, long-term toxicity/safety, and patient consent to treatment. Hence, further exploration of GNPs through preclinical and clinical trials are integral for elucidating the biological mechanisms and effects in order to for clinical usage and to improve targeting to hypoxic tumour microregions relative to normal tissue.

### 2.3. Macrophage-Mediated Drug Delivery: HAPs

#### 2.3.1. Background

Aside from radiosensitizing drugs, alternative drugs classified as hypoxia-activated prodrugs (HAPs) were designed to preferentially kill hypoxic tumour cells through the generation of free radicals [89,90]. The modern approach using macrophages as carriers to deliver HAPs is outlined in Figure 4. The most notable HAP is Tirapazamine (TPZ), which is classified as a benzotriazine-di-*N*-oxide and has shown compelling results both in vitro and in vivo [89]. However, TPZ has not shown any significant results from clinical studies due to physical concerns, such as severe muscle cramping and nausea [91]. In addition, HAPs may potentially enter normoxic cells and elicit negative effects, demonstrating the inefficient selective nature of HAPs [92,93]. Similar to hypoxic cell radiosensitizers, in order to overcome the barriers associated with HAPs, researchers have begun to explore drug delivery systems that integrate nanotechnology and cellular manipulation [94]. Particularly, researchers have been investigating methods to improve the delivery of HAPs to hypoxic regions of a tumour due to their inefficient selective capability through macrophage-mediated delivery systems [94]. Macrophages serve as an attractive vessel for delivering HAPs to hypoxic tumour microenvironments relative to typical drug administration for three main reasons [95]. One reason includes the ability of macrophages to migrate to hypoxic regions of a tumour through chemoattractant gradients [95,96,97,98]. Additionally, macrophages are able to recognize and clear foreign bodies within the bloodstream, which indicates the ability to uptake nanoparticles; thirdly, macrophages have the ability to target various diseases such as cancer [95,96,97,98]. Furthermore, Yu et al. [95] state that macrophages accumulate within hypoxic regions and are activated by intracellular conditions leading to the release of the contents being withheld, such as HAPs. Thus, the innate ability for macrophages to target hypoxic tumour microenvironments relative to normal tissue, demonstrates a powerful approach to combating tumour hypoxia. Favourable biological structural components in relation to drug delivery are outlined further in this paper.

#### 2.3.2. Mechanism of Action

As stated earlier, current research has been focused on advancing drug delivery of HAPs to hypoxic tumour regions mediated by macrophages due to the possibility of HAPs entering normoxic cells and causing detrimental effects on the cell. Macrophages are responsible for the production of inflammatory and antimicrobial cytokines, as well as the removal of pathogens, and are the predominant phagocyte within the immune system [99]. Depending on the environment and host, macrophage regulation can elicit two distinct phenotypes, each with various functionality, M1 macrophages and M2 macrophages, which are further subdivided into M2a, M2b, M2c, and M2d [99]. M1 macrophages are classically activated and generate inflammatory cytokines that inhibit growth, namely, TNF-α and interleukin (IL)-1 [99]. Contrastingly, M2 macrophages are not activated like M1 macrophages and produce anti-inflammatory cytokines and serve as tumour growth promoters [99]. In addition, the four subdivisions of M2 macrophages exhibit various levels of transcriptional changes depending on the stimuli applied [99]. Furthermore, macrophages have the ability to shift between M1 or M2 phenotypes depending on environmental cues [99,100,101]. In the non-targeted field, this was recognized early on [102,103], and it was also found that there was an important correlation between the M2 phenotype, which is correlated with progression to genomic instability and a radioresistant pattern of response, while M1 phenotype is correlated with an apoptotic response to radiation and with a radiosensitive response [104,105,106]. Consideration of NTE in this context might improve treatment outcomes due to a more complete understanding of the mechanisms involved. Interestingly, various diseases, including cancer, demonstrate a disproportionate amount of M1 and M2 macrophage populations [99]. However, within tumours, macrophages are known as tumour-associated macrophages (TAMs), which constitute roughly half of the immune cell populations in tumours and are active during different stages of tumour progression [99,107]. Despite the various phenotype demonstrated by TAMs, overall, these groups of macrophages are classified as M2 macrophages due to the similar responses elicited, such as generating anti-inflammatory cytokines, aids in tumour development, angiogenesis, metastasis, and suppressing the immune system [99,108,109]. Zhang et al. [110] suggest that increased numbers of TAMs, in addition to a high M2:M1 ratio, leads to poor outcomes for various types of cancer. In relation to the poor vasculature system developed by growing tumours, inevitably leading to tumour hypoxia, studies have displayed that TAMs accumulate within hypoxic tumour microenvironments [99]. The migration and infiltration of TAMs are mediated by VEGF and HIF-1. Once entered into the hypoxic region, low intracellular oxygen levels lead to the down-regulation of C-C chemokines, thus rendering TAMs immobile in hypoxic regions [99].

In order to use macrophage-mediated delivery for HAPs, the process requires four distinct phases; “cargo” loading (i.e., whereby cargo refers to the material acquired by macrophages, such as HAPs), maintaining cargo integrity, motility of macrophage in vivo to the target site, and cargo expulsion [94]. In order for macrophages to uptake HAPs, a protective nanoparticle must be encapsulating the drug to prevent the degradation of the drug from intracellular enzymatic conditions induced by macrophages and to protect the macrophage from the drug [99]. For macrophages to uptake a nanoparticle encasing a drug, several characteristics of the nanoparticle surface must be considered for adsorption to macrophage proteins [111]. The three main nanoparticle attributes include curvature, topography, and surface energy; however, other attributes may exist but have yet to be uncovered [111]. Moreover, considerations regarding the ability of macrophage receptors, recognizing these nanoparticles and mechanism of uptake is critical for the drug delivery process [111]. Following uptake of the nanoparticle encasing the drug, it is critical to maintain the integrity of the drug. The drug stability depends on intracellular traffic, particularly the avoidance of lysosomes, due to the potential degradation of the encased drug [112,113]. To prevent cargo degradation, Batrakova et al. [112] suggest that positively charged block-copolymer inhibits lysosome degradation, thus maintaining the stability of the drug. Alternatively, the “backpack” approach is an extreme method for preserving intracellular drug stability, whereby the drug is attached to the surface of a cell carrier [112]. Despite the potential for maintaining drug stability, Batrakova et al. [112] suggest several restraints associated with the “backpack” method, including reduced drug loading, impaired drug release at the target site, and an increase in toxicity and immunogenicity. Consequently, the migration of macrophages is orchestrated through the innate homing properties of macrophages that allow them to travel to hypoxic microenvironments of a tumour [112]. Although the mechanisms related to drug unloading within macrophages continue to be uncovered, several pathways have been hypothesized [112]. One hypothesis suggests that increased concentrations of intracellular calcium is thought to trigger drug release from macrophages [112,114].

Following the release of HAPs from a macrophage carrier, HAPs undergo a series of chemical reactions to activate the drug. Moreover, HAPs display inefficient selective behaviour, which is apparent in one of the most extensively studied HAPs, TPZ [115]. In principle, HAPs are masked or deactivated cytotoxins that are subjected to biotransformations, which are then proceeded by reductive metabolism orchestrated by intracellular oxidoreductases to produce an active compound [116]. Initially, compounds remain inactive due to the positioning of a bioreductive protecting group, which is released following reduction and fragmentation [117,118]. As described by Guise et al. [116], normally, the aforementioned processes are inhibited within normoxic cells due to the levels of intracellular oxygen. However, the significantly lower oxygen concentration is ideal for HAP activation within hypoxic tumour cells [116]. Furthermore, the activation of HAPs can occur in two different manners, through one-electron oxidoreductases to catalyze oxygen-sensitive HAPs or through two-electron oxidoreductases to catalyze oxygen-insensitive HAPs [116]. In regard to oxygen-sensitive HAPs, one-electron oxidoreductases will produce free radicals that can easily be reoxidized into the inactive precursor form, creating a futile metabolic cycle, thus, limiting these HAPs to hypoxic regions [116,119]. Contrastingly, as described by Guise et al. [116], processes using two-electron oxidoreductases are irreversible and are unable to produce oxygen-sensitive radical intermediates; thus, the compound can potentially situate within normoxic and hypoxic tissues, overall, creating HAP activation independent of oxygen concentration. Despite the role of one-electron oxidoreductases and two-electron oxidoreductases in relation to HAP activation, the expression and frequency of these enzymes remain to be elucidated within human tumours [116]. Reduction of TPZ can either be performed by one-electron or two-electron oxidoreductases; however, Phillips [120] states that reduction by one-electron oxidoreductases, cytochrome P450 reductase (P450R), is the primary pathway. Despite extensive clinical trials conducted in the early 2000s examining the potential of TPZ [121,122,123,124], no significant results have been produced in relation to treating cancer. Moreover, recent research has been exploring the use of HAPs, namely, TPZ, mediated through a macrophage-mediated drug delivery system in order to overcome the ineffective selectivity behaviour of TPZ. Thus, the integration of TPZ within macrophage-mediated delivery systems poses as an attractive method to target hypoxic tumour sites relative to normal tissue.

#### 2.3.3. Current Studies and Future Developments

Although several HAPs, most notably, TPZ, have successfully advanced to clinical trials, several shortcomings have prevented the approval of HAP in clinical settings [125]. Recent studies have focused on overcoming the drawbacks of HAPs using nanotechnology and cellular manipulation to develop mechanisms for macrophage-mediated drug delivery of HAPs. Particularly, Evans and colleagues [126] have demonstrated the potential use of macrophages through in vitro and in vivo analyses. Through both experiments, a hydrophobic derivative of TPZ encapsulated within poly(lactic-co-glycolic) acid (PLGA) nanoparticles are contained within a macrophage and are collectively referred to as MAC-TPZ [126]. Prior to conducting in vivo analyses, Evans and colleagues [126] orchestrated multiple in vitro assays to confirm the ability of macrophages to uptake TPZ. Furthermore, in vitro analyses allowed for comparing different combination lengths of carbon chains attached to TPZ in order to determine effects on nanoparticle properties, toxicity, and overall efficacy on treatment outcomes [126]. Following the series of in vitro analyses, it was determined that TPZ-C12 proved to be the most optimal of the combinations analyzed, due to the balance between hydrophobicity and hydrophilicity, thus, preventing premature macrophage toxicity and efficient diffusion within the tumour, respectively [126]. Consequently, the biological confirmation of macrophage drug uptake and the optimal drug used within the macrophages are collectively referred to as MAC-TPZ_C12_. Evans and colleagues [126] conducted in vivo analyses were conducted on mice injected with a type of breast cancer cell (4T1). Subsequently, the group [126] concluded that mice treated with MAC-TPZ_C12_ displayed a 3.7 reduction in tumour size and weight, relative to mice treated solely with TPZ and TPZ-C12.

Overall, this study provides compelling evidence for the use of macrophage-mediated drug delivery systems for HAPs as a novel technique for treating tumour hypoxia. In addition, the conclusions drawn from this study strongly suggest that one of the major shortcomings of TPZ, the inability to effectively target hypoxic tumour microenvironments relative to normal tissues, can be overcome through macrophage-mediated drug delivery. Although this technique has only recently emerged, this study serves as an exciting gateway for exploring extensions of this technique to other HAPs or chemotherapeutic drugs and other types of cancers. Unsurprisingly, as this technique is still in its early developments, much more extensive research through preclinical trials is required prior to the translation to clinical settings.

### 2.4. Autophagy and Tumour Metabolism

#### 2.4.1. Background

Another method is the use of drugs that target tumour metabolism. Tumours demonstrate a growth advantage through a shift in metabolism, specifically, from oxidative phosphorylation to glycolytic metabolism, which is driven by HIF-1–pyruvate dehydrogenase kinase 1 (PDK1) [127]. Consequently, the shift in metabolism causes tumours to conserve oxygen supplies and induces a compensatory response by increasing glycolysis through the reduction of mitochondrial processes [127]. Moreover, the inhibition of PDK leads to an increase in tumour hypoxia [127]. Researchers have demonstrated that dichloroacetate (DCA) acts as a PDK inhibitor, increasing mitochondrial functioning within tumours, thus, resuming oxidative metabolism similar to normal tissues [128]. Aside from targeting the transition from oxidative phosphorylation to glycolytic metabolism, other cancer therapeutics targeting autophagy, another molecular process aiding in tumour metabolism, has been garnering recent attention. The overall concept for this approach is outlined in Figure 5. The process of autophagy in relation to cancer involves the degradation of damaged cellular components that are recycled to meet the metabolic demands of cancer cells [129]. Mizushima and Komatsu [130] suggest that low baseline levels of autophagy are essential for preventing toxicity in tissues by preventing the build-up of damaged proteins and organelles. Autophagy is characterized as a “double-edged sword”, primarily due to its dual role in tumourigenesis serving as a tumour suppressor and tumour promotor, depending on the type of tissue and stage of the tumour [129]. Several studies on human prostate, breast, and ovarian cancers, displayed partial monoallelic loss in one essential autophagy gene, ATG6/Beclin-1 [129,131,132,133]. Furthermore, the impairment of proper autophagy functioning in tumours serves as a signal for identifying cancer [134]. Research has also suggested that autophagy acts as a tumour promoter due to growth enhancement and survival capabilities [135]. As previously mentioned, hypoxic tumour environments are severely lacking in metabolic requirements due to primitive vasculature systems; however, autophagy serves to meet metabolic demands through the recycling of intracellular components [135,136,137]. Normally, when damaged or old cells are removed through autophagy, there is a release of structural biological components, such as amino acids, nucleotides, and fatty acids [138]. Furthermore, these intracellular components can be recycled and used for tumour metabolic demands; however, suppressing autophagy through the partial deletion of the Beclin-1 gene leads to increased cell death [135]. Current preclinical studies have determined that inhibiting autophagy has improved cancer patient outcomes [139]. At present, the only autophagy inhibiting drugs viable for clinical studies are chloroquine (CQ) and hydroxychloroquine (HCQ), a derivative of CQ.

#### 2.4.2. Mechanisms of Action

As previously mentioned, recent studies on tumour metabolism have evolved from analyzing the shift in oxidative to glycolytic processes within tumours to the role of autophagy in tumour progression and the subsequent development of autophagy inhibitors. In particular, the most notable autophagy inhibitors are CQ and HCQ, which are classified as 4-aminoquinoline agents and were first intended as anti-malarial drugs [138]. HCQ is a derivative of CQ and is distinguished by the addition of a hydroxyl group on the beta carbon of the tertiary amino ethyl situated on the terminus side of the quinolone base [140]. Furthermore, the addition of hydroxyl group restricts the movement of HCQ across blood-retinal barriers, overall resulting in lower toxicity relative to CQ [138,140,141]. As stated earlier, autophagy has two distinct roles in relation to tumourigenesis, namely as a tumour suppressor or tumour promoter depending on the stage of tumour development and the tissue type [138]. During the early stage of tumour development, autophagy acts as a tumour suppressor due to its ability to clear defective cells, thus, maintaining cell homeostasis [138,142,143]. Moreover, various proteins associated with autophagy that directly suppress tumour development include Beclin-1, UVRAG, and Bif-1, as well as components that destroy proteins associated with tumour growth such as p62/SQSTM1 [138,144]. Contrastingly, during the later stages of tumourigenesis, levels of autophagy increase and act as a tumour promoter in response to harsh intracellular environments such as starvation, hypoxia, and organelle damage [138]. Furthermore, one major characteristic of autophagy is the ability to recycle nutrients, which can be used to sustain tumour development [138]. Moreover, increased autophagy activity is associated with the destruction of cell growth regulators, as well as suppression of DNA damage mechanisms [138,142,143,144]. In order to combat autophagy, CQ and HCQ act as inhibitors during the late stages of autophagy, particularly when lysosomes and autophagosomes fuse together [138]. Townsend et al. [140] state that when CQ or HCQ enter the lysosome, this causes the protonation of these compounds, ultimately trapping these compounds within the acidic lysosome environment, causing the inhibition of lysosome degradation enzymes. Lysosomes are integral during autophagy, as these cells are responsible for the degradation of macromolecules that can be reused within cells [145]. Thus, cells treated with CQ or HCQ cannot undergo lysosomal digestion [140,146]. Moreover, preventing the proper lysosome functioning, in turn, prevents the supply of macromolecules required for tumour growth, thus, serving as an attractive method for targeting tumour metabolism relative to normal tissue [138].

#### 2.4.3. Possible Involvement of Non-Targeted Effects

The mechanism of autophagy and its relationship to radiation and NTE has been well studied [147,148,149]. Bystander effects have been known since the early 2000′s to involve mitochondria and to be modulated by both anaerobic and aerobic metabolism and metabolic inhibitors [150,151]. Several recent studies have shown rescue effects, whereby signals from non-targeted cells can alter the fate of directly irradiated cells [152,153]; although the differences between normal and tumour tissue responses involving autophagy have been alluded to, no studies have been performed looking at autophagy in relation to bystander or other NTE in hypoxic cells. It is likely there will be very complex interactions involved, and it is clearly an important area to investigate to progress the development of hypoxic cell modulators, which involve autophagy.

#### 2.4.4. Current Studies and Future Developments

As recent research has shifted away from tumour metabolism to autophagy, many preclinical and clinical trials are examining the use of autophagy inhibitors, namely CQ and HCQ, to treat tumour hypoxia. Due to the extensive clinical trials conducted on CQ and HCQ, this review will focus on the early clinical trials and current developments. Early clinical trials conducted by Briceño and colleagues [154] examined the effect of radiation with CQ and an alkylating agent known as temozolomide within a small group of patients with glioblastoma. From this study, it was concluded that individuals who are part of the treatment group demonstrated a better prognosis, evident through a median survival of 33 months relative to those who were not treated and had a median survival of 11 months [154]. Consequently, Briceño et al. [154,155] and Sotelo et al. [156] conducted follow-up studies on the clinical trials and validated their findings from the clinical trials that were first conducted. Similarly, more recent experiments conducted by several groups have examined HCQ in combination with existing chemotherapeutic agents such as temozolomide [157], bortezomib [158], temsirolimus [159], and vorinostat [160]. Overall, these studies demonstrated that patients with melanoma, colorectal cancer, myeloma, and renal cell cancer, responded favourably to treatment with HCQ, which indicates the potential of HCQ as an anticancer therapeutic drug [157,158,159,160,161].

However, one of the major shortcomings of solely using CQ or HCQ is the inability to determine changes in autophagy, such as increases or decreases in activity, termed as “autophagy flux”; hence, current research is aimed at identifying autophagy biomarkers that detect these levels [139,162]. Notably, a study conducted by Barnard et al. [163] determined that there is a link between HCQ and autophagy flux inhibition and was evident through the formation of LC3II and sequestosome 1, which were identified as potential autophagy biomarkers. As mentioned previously, autophagy is regarded as a “double-edged sword” due to its dual effect on tumourigenesis; however, this characterization is also attributed to CQ and HCQ. Despite the effective results of CQ and HCQ as an autophagy inhibitor in order to suppress tumourigenesis, autophagy is still an essential process within normal tissue [164]. The homeostatic nature of this process within normal cells may be disrupted by these drugs and can potentially cause additional issues, such as acute/chronic toxicities to organs, most notably on the kidneys [164]. The kidneys are most likely to be affected by these drugs since many chemotherapeutic agents and metabolites are excreted from kidney tubular epithelial cells [164,165]. Kimura and colleagues [164] further state that using CQ as an autophagy inhibitor can potentially lead to the sensitization of kidney cells to anticancer treatments, such as radiotherapy, resulting in acute kidney injury.

Although CQ and HCQ are currently the only autophagy inhibitors being investigated for clinical use, future objectives should be aimed at uncovering potentially more potent and efficient autophagy inhibitor drugs. In addition, the cellular mechanisms and relationship between autophagy biomarkers, such as LC3II and sequestosome 1, have yet to be completely elucidated. Hence, additional research must be conducted to examine currently known autophagy biomarkers, as well as identifying novel biomarkers in relation to responses to CQ, HCQ, and potentially novel autophagy inhibitor drugs. Despite the potential for developing acute kidney injury resulting from CQ-based anticancer therapies, validating the cause-and-effect relationship may not be possible [164]. The uncertainty regarding the causal relationship is attributed to the difficulty in monitoring autophagy in human biopsy specimens. Even if it is feasible, this information alone might not be sufficient for confirming the effect of CQ [164,166]. Furthermore, long-term studies analyzing the effect of CQ and the development of acute kidney injury are required in order to establish CQ and HCQ as viable treatments for tumour hypoxia. Overall, CQ and HCQ display promising results as chemotherapeutic agents targeting tumour metabolism through autophagy. However, the major conflict lies between successful clinical trials thus far and the potential of acute kidney injury due to prolonged usage. These effects must be extensively investigated prior to clinical translation.

### 2.5. Non-Targeted Effects as a “Target”

#### 2.5.1. Background

While most work aiming to improve the therapeutic ratio focuses on sensitizing the tumour tissue, an equally valid approach is to increase the resistance of normal tissue. One approach to this is to manipulate bystander signaling. The bystander effect refers to effects in cells that were not themselves irradiated but that were in receipt of signals from irradiated cells [167]. Such signals induce a range of responses, including death of cells [168], induction of genomic instability [169,170,171], mitochondrial changes and other effects [172,173], which appear to result from increased ROS in the responding cell [174,175]. Most of these effects make the normal unirradiated cells more sensitive to death and may account for out-of-field and memory or legacy effects [176,177,178]. However, there have been reports of survival enhancing bystander effects [179], and induction of adaptive responses by bystander signals have also been reported [180,181,182]. Since NTE saturate at low doses of approximately −0.5 Gy [183,184] and are induced by mGy doses [185,186], they could potentially impact the survival of surrounding normal tissue as well as normal cells within the tumour (e.g., fibroblast or endothelial cells). There are strong reasons to suspect that hypoxic cells or those with compromised oxidative metabolism will have reduced or absent cytotoxic bystander effects. This comes from several lines of evidence. We have known for many years that persistent oxidative stress is induced in recipients of bystander medium [187,188] and has also been linked to the induction of genomic instability in both directly irradiated and bystander cells [189]. Experiments using cell lines with mitochondrial malfunction (glucose-6-phosphate dehydrogenase deficient) do not show a cytotoxic bystander effect [150]. Many tumour cell lines that respire anaerobically do not show cytotoxic bystander effects [190,191]. However, it is important to stress that the absence of reduced cloning efficiency, micronucleus, or chromosomal damage endpoints in bystander experiments does not mean that no signal or effect was produced. It merely means that the response of the cells was not among the endpoints examined. The important point for hypoxic tumour radiotherapy is that it is likely that normal well-oxygenated cells will experience more cell death owing to bystander-related mechanisms than hypoxic, anaerobically respiring tumour cells. Considering this might suggest new therapeutic approaches to improve the therapeutic ratio, as indeed the recent paper by Zhang et al. [192] confirms. The key to this is whether the signal production and response in the recipient are capable of being modulated independently. Several papers over the last 20 years suggest they are [17,193]. Mix and match media exchange experiments in which the media from a producer of the signal is added to a cell line that does not show a bystander effect and vice versa, confirm that non-responding cells can produce a robust effect in a known responding cell line, but when media from a cell line that does not produce a signal is added to a recipient that can respond, a smaller bystander effect is seen [16,194].

#### 2.5.2. Non-Targeted Effects Mechanisms

The aim of manipulating NTE would be either to turn the mechanism ON in hypoxic tumour cells or OFF in normal cells. Identifying ways to do this requires detailed knowledge of how NTE are initiated and perpetuated in cells. Figure 6 shows our current understanding of NTE and identifies potential intervention points where there are differences in the response of normal cells and tumour cells. The key points are divided into those that could prevent signal generation and those that prevent the response of the cell receiving a signal. Among the most promising intervention points is that involving inhibition of 5-hydroxy tryptamine (5HT) binding by the 5HT receptors on the cell membrane. These receptors are ion-gated and control the entry of calcium into the cell [195,196,197,198]. Intracellular calcium ion increase is one of the first signs when a bystander signal is received, occurring within 30 s of receipt [199]. This target is especially promising since 5HT receptor inhibitors such as ondansetron are already used during radiotherapy to alleviate radiation-associated emesis [200]. Another molecular target could be p53. This tumour suppressor gene is mutated in many cancers, and in others, it has a stable conformation meaning it remains in the cytoplasm and is effectively non-functional [201]. Our studies have shown that normal wild-type p53 protein function is required for cells to respond to bystander signals [17,193]. Compromised protein function means cells can still produce signals, but they cannot respond. This results in normal tissue damage occurring as a result of signals produced by tumour cells and presents an unfavourable therapeutic ratio. It is possible that gene therapy approaches might work to provide wild-type p53 to tumour cells allowing bystander signalling to trigger apoptosis. Recent studies have indicated an important role for exosomes in the communication of bystander signals. These appear to be released by directly irradiated cells and to be taken up by neighbouring (or distant?) unirradiated cells [193,202,203]. MiRNAs, as well as proteins, have been identified as important contents of the exosome, and it has recently been shown that exosomes harvested from irradiated cells can induce bystander effects if isolated and added to unirradiated cultures of cells [204,205]. It is unlikely that exosomes could trigger apoptosis in p53 incompetent tumours, but in tumours that have fully functional p53, supplying exosomes might be effective in turning on bystander responses. NTE are known to be triggered by oxidative stress possibly generated by the recently demonstrated emission of UVA photons by directly irradiated cells [206,207,208]. A key reason why hypoxic cells are thought to be resistant to bystander signals is that in the absence of oxygen, toxic ROS cannot be generated [209,210]. This suggests that using antioxidants as radioprotectors would act to improve the therapeutic ratio and reduce normal tissue damage.

#### 2.5.3. Current Studies and Future Developments

Except for one very recent paper from Zhang et al. [192], who found that increased micronucleus formation and decreased survival were seen in cells cultured under normoxic or hypoxic conditions when treated with medium harvested from irradiated cells or co-cultured with irradiated cells, there has been very little research on radiation-induced bystander effects in hypoxic and normoxic conditions. The paper by Zhang suggests a complex reaction in hypoxic tumour cells, where HIF-1α expression was increased, suggesting a role for this protein in regulating bystander response in hypoxic cells. There has also been little research that we could find into whether hypoxic cell sensitizers could enhance bystander effects in hypoxic areas of the tumour. There is considerable literature suggesting that certain novel radiotherapy regimes could be deliberately set up to induce non-targeted effects. Massaccesi et al. [211] suggested the possibility of intentionally triggering NTE by using spatially fractionated radiotherapy (SFRT), possibly in combination with immunotherapy, to target the hypoxic parts of the tumour. They produced an in silico model supporting their suggestion. It is not clear, however, what cells in the tumour ecosystem would develop the NTE. A similar experimental approach was actually tried out by Tubin et al. [212]. They did a phase two trial of stereotactic body radiotherapy targeting partially hypoxic clonogenic tumour cells in non-small cell lung cancer patients and found significant bystander effects, which they attribute to the sparing of the immune-competent microenvironment of the peripheral areas of the tumour. However, since immune response elements (e.g., cytokine signaling) are part of the response to bystander signals, these approaches will only work if the hypoxic tumour cells are producing the initial signals. We know many tumour cell lines do, but these are not usually cultured under hypoxic conditions. The paper discussed above [192] appears to be the only research specifically addressing this aspect of hypoxic tumour radiobiology. This is clearly an area needing more research.

A very different set of papers in the hypoxia literature uses the term “bystander effect” (e.g., see [213,214,215]). These papers are mentioned here to demonstrate the unique difference between NTE and the use of the term “bystander effect” in another context. Foehrenbacher et al. [213] and Hong et al. [214,215] use the term to describe the diffusion of drugs activated by the hypoxic environment into the microenvironment of the tumour where normal and normoxic cells occur. They are mainly concerned that such diffusion could cause collateral damage to normal cells, adversely affecting the therapeutic ratio. This is completely different from the active process of radiation-induced bystander effects, which involve complex signaling pathways.

## 3. Conclusions

It is widely understood that tumour hypoxia has proved to be a major barrier for treating the majority of solid tumours. In response to this longstanding problem, despite extensive research on early tumour hypoxia targeted therapeutics, these treatments have faced several limitations, including the ineffective targeting to hypoxic microregions relative to normal tissue. This complex issue led researchers to integrate novel technological advancements, such as nanotechnology, medical imaging, and cell manipulation to create novel approaches. Furthermore, the shift in perspectives led to the development of modern approaches including, imaging using PET to mediate carbogen breathing, GNPs, macrophage-mediated drug delivery for HAPs, and indirectly targeting tumour metabolism through autophagy inhibitors. All four novel approaches aim to improve anticancer therapeutics, primarily radiotherapy outcomes, as well as increasing the efficacy of targeting hypoxic tumour microenvironments relative to normal tissue. These therapeutics are still within the early stages of development, some reaching preclinical trials such as carbogen breathing guided by PET tumour hypoxia imaging and macrophage-mediated drug delivery, while others have reached clinical trials and await clinical translation, such as GNPs and CQ/HCQ. Furthermore, the knowledge affiliated with these novel therapeutics has seen a surge in attention; however, the biological mechanisms and the road to clinical translation have yet to be completely elucidated. Thus, in order to further potentiate the use of these approaches as novel tumour hypoxia therapeutics, thorough investigations must continue. Each of these novel approaches may be impacted differently by the presence of NTE, and several ways that NTE could be harnessed to improve the therapeutic ratio are discussed; however, this review found little evidence that this approach is being considered, and we suggest it is a novel and important avenue for research in the future.

This review was centered around outlining the evolution of the methods for radiosensitizing hypoxic tumour cells and improving the targeting of hypoxic tumour microregions relative to normal tissue. Moreover, this review provides a comprehensive summary of the molecular mechanisms that allow for enhanced targeting and radiosensitization of hypoxic tumour microenvironments relative to normal tissue, preclinical/clinical studies examining the efficacy, and future considerations for clinical translations.

## Figures and Tables

**Figure 1 ijms-22-08651-f001:**
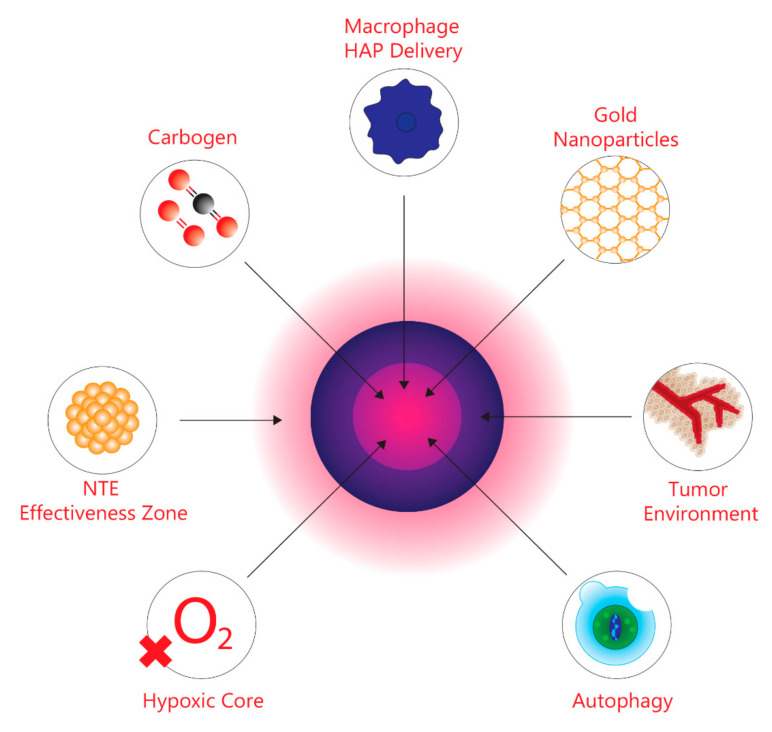
Graphic depicting the tumour microenvironment and the targets discussed in this review.

**Figure 2 ijms-22-08651-f002:**
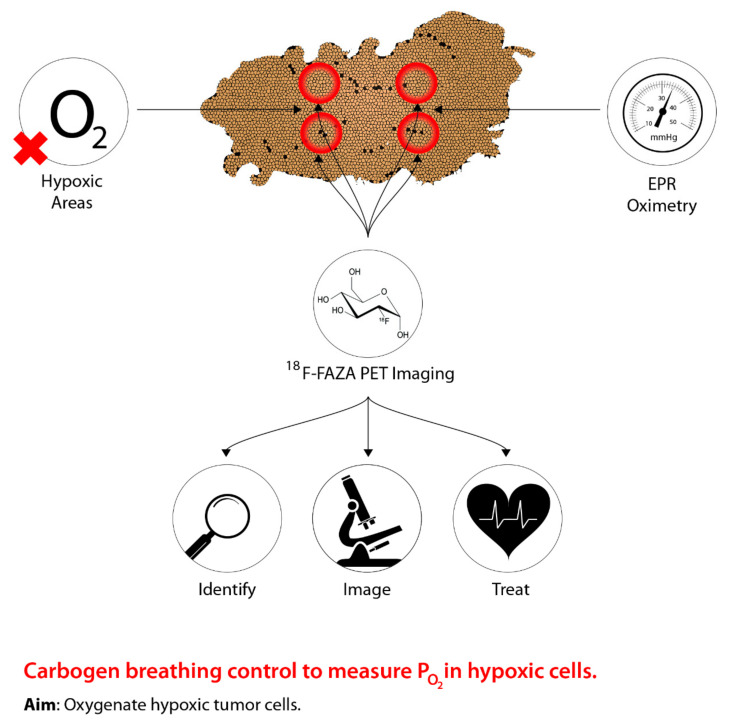
The concept underlying PET imaging to optimize carbogen breathing approaches to defeating hypoxia in radiotherapy. Oxygen levels are measured as the tumour is imaged, thus allowing precise identification of hypoxic regions, which can then be effectively targeted.

**Figure 3 ijms-22-08651-f003:**
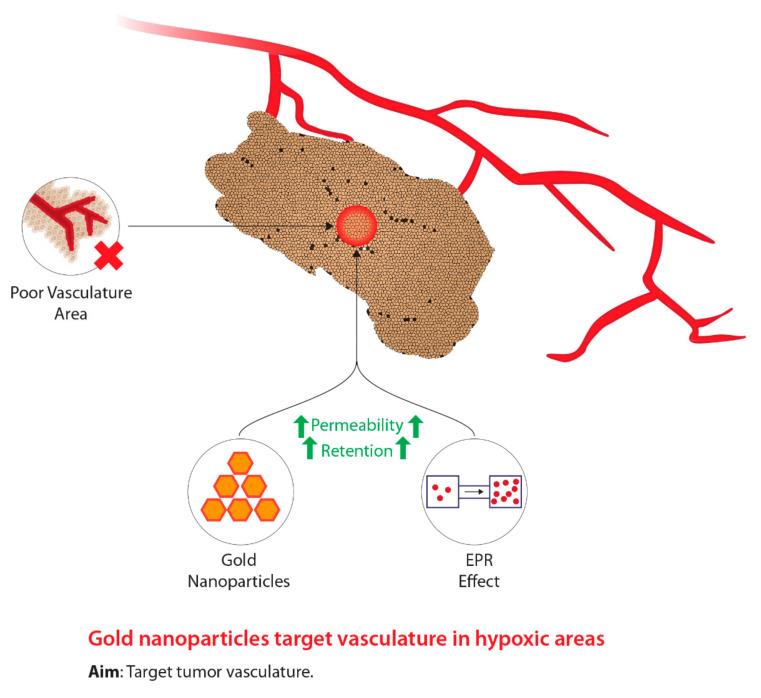
The concept underlying the use of gold nanoparticles in the treatment of tumour hypoxia during radiotherapy.

**Figure 4 ijms-22-08651-f004:**
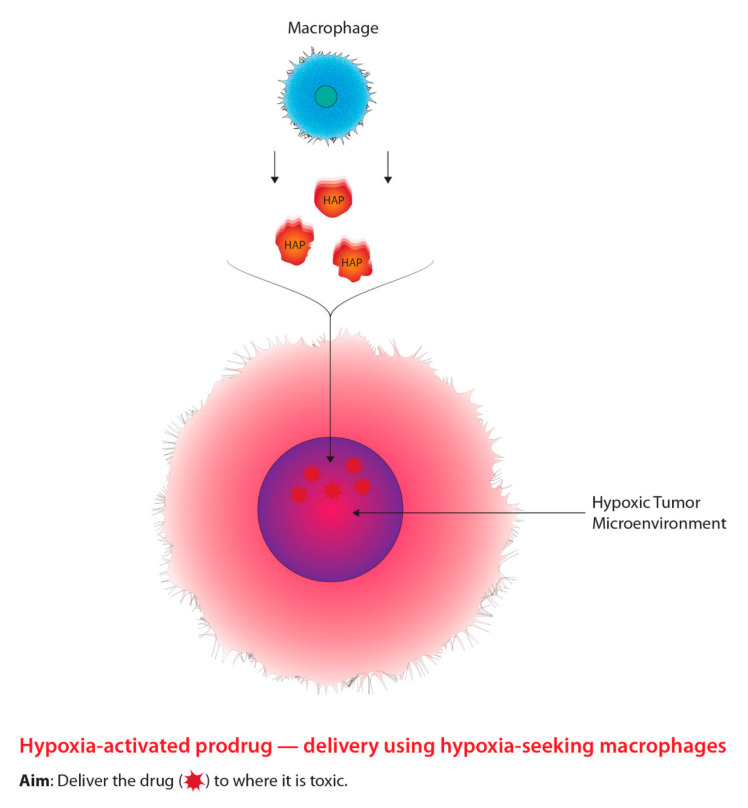
The concept underlying the targeting of hypoxic cells using prodrugs tagged to tumour-seeking macrophages.

**Figure 5 ijms-22-08651-f005:**
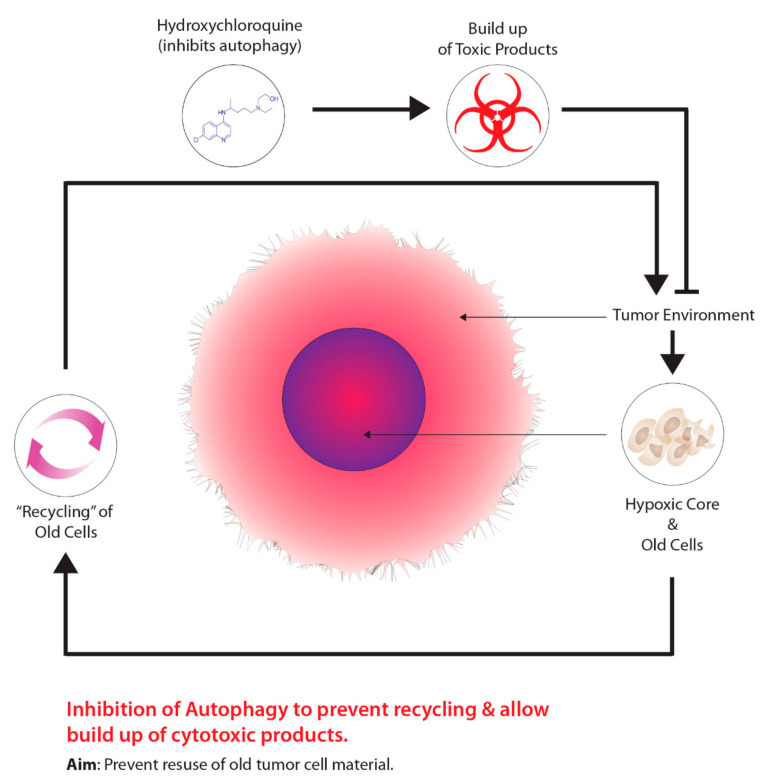
Harnessing autophagy to destroy hypoxic cells.

**Figure 6 ijms-22-08651-f006:**
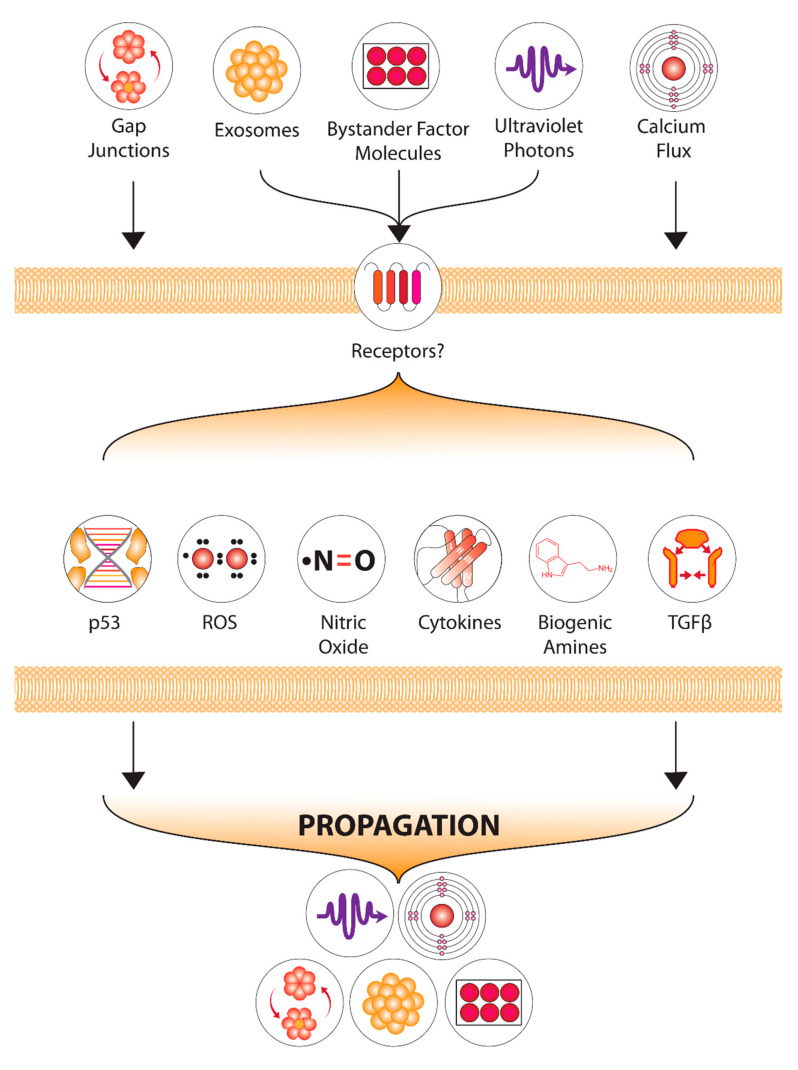
Current understanding of non-targeted effect mechanisms with possible intervention points to optimize hypoxic tumour cell kill.

## Data Availability

Not applicable.

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
