# Peer review of "Targeted and Non-Targeted Mechanisms for Killing Hypoxic Tumour Cells—Are There New Avenues for Treatment?"

_ijms, 2021, doi:10.3390/ijms22168651_

Round 1
Reviewer 1 Report
This review provides a comprehensive summary of the current state of play in cancer therapeutics exploiting tumor hypoxia. The review is well organised and although somewhat long for the messages it contains, will be a useful source of reference for the field, and is a useful and novel overview regarding bystander effects in this area. The problem I found was that in many places the english was idiosyncratic or contained grammatical or syntactical errors; these should be carefully edited. The only suggestion I have for a change in content is that the mechanism of the apparently paradoxical application of hyperbaric oxygen should be explained. The number of places that need editorial attention is such that I have highlighted them on the PDF and attach to this review for the authors benefit.

Reviewer 2 Report
none
Author Response
Thank you for the positive feedback.